# SemanticMIM: Marring Masked Image Modeling with Semantics Compression for General Visual Representation

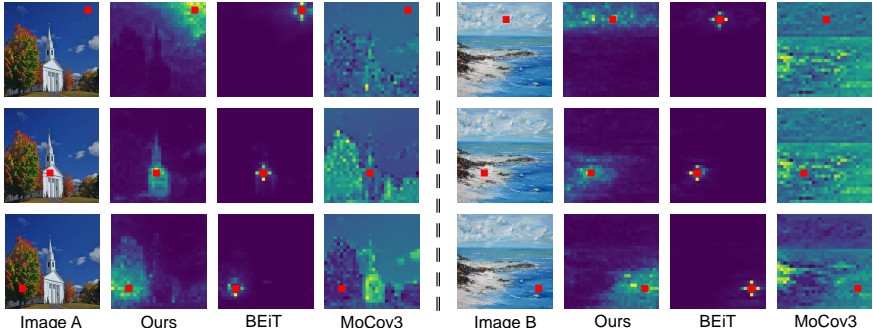

| Image A | Ours | BEiT | MoCov3 | Image B | Ours | BEiT | MoCov3 |

Figure 1: Attention response of different self-supervised vision transformers. The queries are marked with red boxes. MoCov3 fails to follow the query and BEiT focuses too much on neighboring patches, while SemanticMIM distinguishes different objects and approximates their segmentation masks. MoCov3 and BEiT show the result from $10^{th}$ layer while Ours are from $8^{th}$ layer. Attention responses across depth are further analyzed in supplementary.

## Abstract

This paper represents a neat yet effective framework, named SemanticMIM, to integrate the advantages of masked image modeling (MIM) and contrastive learning (CL) for general visual representation. We conduct a thorough comparative analysis between CL and MIM, revealing that their complementary advantages may stem from two distinct phases, *i.e.,* compression and reconstruction. Specifically, SemanticMIM leverages a proxy architecture that customizes interaction between image and mask tokens, bridging these two phases to achieve general visual representation with both abundant semantic and positional awareness. Through extensive qualitative and quantitative evaluations, we demonstrate that SemanticMIM effectively amalgamates the benefits of CL and MIM, leading to significant enhancement of performance and feature linear separability, and also offers notable interpretability through attention response visualization.

## 1 Introduction

Self-supervised learning (SSL) algorithms (Liu et al., 2021; Balestriero et al., 2023) have emerged as a powerful paradigm for deriving rich feature representations without relying on extensive annotations. These algorithms can be roughly categorized into two families: Masked Image Modeling (MIM) (He et al., 2022; Xie et al., 2022) and Contrastive Learning (CL) (He et al., 2020; Chen et al., 2020a). As illustrated in Fig. 1, MIM focuses on the reconstruction of partially corrupted images, serving as a pretext task that facilitates the model's ability to infer local patterns from limited contextual information, however the redundancy of image signals hinders the learning of grasping long-range global semantics (Li et al., 2023; Xie et al., 2023b). MIM is inherently compatible with the transformer architecture and demonstrates versatility across different tasks and modalities, thereby garnering increasing research interest. In contrast, Contrastive Learning emphasizes aligning global

features with instance discrimination as its core pretext task (Wu et al., 2018). CL excels in capturing prominent, semantically rich foreground elements, albeit at the expense of nuanced understanding of complex local spatial patterns. Further, the absence of positional priors in the pre-training implies that CL's semantic understanding is broadly homogeneous, circumventing the need for explicit positional awareness. Consequently, MIM and CL exhibit specialization in downstream tasks that are sensitive to positional dynamics (*e.g.,* segmentation) and semantic content (*e.g.,* classification), respectively. Given the distinct properties of MIM and CL, there exists a compelling imperative to find a compromise solution that can absorb the advantages of both methods.

Prior efforts to reconcile the disparities between MIM and CL have predominantly adopted two strategies. First, one approach sought to augment CL with fine-grained alignment with positional priors, such as aligning features of pixels, regions, or objects (Wang et al., 2021; Van Gansbeke et al., 2021; Bai et al., 2022). However, this CL-centric strategy suffers from collapse to trivial solutions and thus heavily relies on hyperparameters and regularization, thereby sacrificing the inherent flexibility of MIM. Second, a more straightforward strategy is optimizing the objectives of MIM and CL simultaneously (Zhou et al., 2021; Oquab et al., 2023). This approach inevitably introduces the complex task of integrating two distinct learning objectives and the pursuit of multi-view representation significantly escalates the computational demands, which necessitates a nuanced approach to balance and fuse these methodologies.

To deepen the understanding of the intrinsic properties of MIM and CL, we explore their specifics empirically in Sec. 3.2. We elucidate that the complementary capabilities of CL and MIM methods in semantic modeling, *i.e.,* consistency and completeness are achieved through *compression* and *reconstruction*, respectively. CL approaches compress information from all image patch tokens `[IMG]` into a single class token `[CLS]`, encapsulating global abstract semantics. Conversely, reconstruction-based MIM methods prioritize local neighbors rather than global semantics, stemming from the inherent redundancy in image modality.

Inspired by CL, we propose SemanticMIM, a novel paradigm that introducing compression within MIM framework, aiming to harness some of the advantages of CL methods. It is noteworthy that SemanticMIM is not a combination of CL and MIM in a multi-task manner. Instead, it strictly adheres to the general MIM framework and achieves compression by controlling information exchange. Specifically, we propose a proxy architecture to seamlessly connect two phases: initially, `[IMG]` tokens interact with `[PROXY]` tokens, compressing all information into the `[PROXY]` tokens, which embody abstract semantics. Subsequently, these `[PROXY]` tokens engage with the mask tokens `[MASK]`, reconstructing the target with spatial priors derived from `[MASK]` tokens while preserving rich semantics through the `[PROXY]` tokens.

We conduct a broad spectrum of both qualitative and quantitative analyses to substantiate the efficacy of SemanticMIM. The following points delineate the advantages of SemanticMIM. (1) Compared to MIM, SemanticMIM excels in discerning the semantics of specific objects rather than semanticless neighbor pixels, *i.e.,* consistency. (2) Compared to CL, SemanticMIM exhibits a keen positional awareness rather than homogeneous perception. It adeptly identifies targeted semantics within both foreground and background elements with explicit positional priors, *i.e.,* completeness. (3) The `[PROXY]` tokens tend to inherently learn various implicit positional priors, *i.e.,* regions of interest. This mechanism naturally directs the model's attention towards relevant semantic features. (4) Beyond achieving considerable performance gains in fine-tuning settings, SemanticMIM significantly improves the performance in linear probing settings, which indicates that the features of pre-training phase are more linearly separable.

The main contributions are summarized as follows:

- We provide an elaborate analysis, and point out that the fundamental principles underlying contrastive learning and masked image modeling lie in compression and reconstruction, respectively.
- We propose SemanticMIM, a neat yet effective framework to integrate the merits of masked image modeling and contrastive learning. SemanticMIM leverages a proxy architecture to orchestrate compression and reconstruction in cascades.
- Extensive qualitative and quantitative experiments validate the effectiveness of SemanticMIM, indicating its capability of obtaining visual representations with high consistency and completeness.

## 2 RELATED WORK

### 2.1 MASKED IMAGE MODELING

Motivated by the success of Masked Language Modeling in NLP (Devlin et al., 2018), Masked Image Modeling (MIM) has been proposed for training a vision transformer in a self-supervised manner. For MIM task, the model takes a corrupted image as input and predicts the target of the missing area. The difference between the prior works of MIM mainly lies in the target choice and image corruption.

Target can be roughly divided into two kinds: low-level signals and high-level features. The former mainly refers to raw pixels (Dosovitskiy et al., 2021; Xie et al., 2022; Huang et al., 2022b), normalized pixels (He et al., 2022), hand-crafted feature descriptors (Wei et al., 2022a), and even positions (Zhai et al., 2022; Caron et al., 2024; Wang et al., 2024). This kind of target is easy to obtain with no extra cost but continuous signals suffer from high redundancy and few semantic information. Researchers find that high-level features extracted by well-trained image tokenizers are also appropriate targets, including concrete deep features by offline (Wei et al., 2022a;b; Zhou et al., 2022; Hou et al., 2022; Peng et al., 2022) or online models (Tao et al., 2023; Chen et al., 2022; Dong et al., 2022; Zhou et al., 2021) and discrete codes (Bao et al., 2022; Chen et al., 2024; Dong et al., 2023) generated by VQ-VAE (Van Den Oord et al., 2017) or dVAE (Ramesh et al., 2021).

The most used image corruption strategy is randomly removing a certain proportion of image patches. On this basis, removing multiple adjacent patches creates a more challenging context and encourages long-range dependency (Bao et al., 2022; Xie et al., 2022). Inspired by hard sample mining, manually designed criterions are proposed to choose where to mask and guide the model reconstructing the discriminative image patches (Kakogeorgiou et al., 2022; Shi et al., 2022; Li et al., 2021). Furthermore, image corruption could also be processed in the frequency domain, transforming the pretext task into low-level vision tasks such as image super-resolution or denoising (Xie et al., 2023a).

### 2.2 CONTRASTIVE LEARNING

Contrastive learning (CL) methods (He et al., 2020; Chen et al., 2020a;c; Chen* et al., 2021; Chen et al., 2020b) learn visual representations by creating different views of an image and aligning their features, encouraging semantic invariance to simple transformations. To resist collapse to trivial solutions, the contrastive loss (Wu et al., 2018; Oord et al., 2018) is adopted to maximize dissimilarity between negative sample pairs. The introduction of prototypes (Li et al., 2020; Asano et al., 2019; Caron et al., 2020) addresses the large batch size requirement by replacing pairwise comparison with cluster assignment consistency. Self-distillation methods (Grill et al., 2020; Caron et al., 2021; Oquab et al., 2023; Chen & He, 2021) further simplify the training framework, preventing collapse by asymmetric model architecture and parameter update strategy. Besides, more regularizations (Zbontar et al., 2021; Bardes et al., 2021; Oquab et al., 2023) are proposed to constrain correlations in the dimensions of not only samples but also features.

On the other hand, CL is sub-optimal for dense prediction downstream tasks due to the discrepancy between image-level alignment pre-training and pixel-level prediction. Hence, dense contrastive learning methods are proposed aligning sub-image-level features with position priors. Pixel-level (point-level) features are easily obtained on feature maps before pooling and matched between views by designed rules such as similarity or spatial distance (Wang et al., 2021; O Pinheiro et al., 2020; Xie et al., 2021b; Zhang et al., 2022; Ziegler & Asano, 2022). With the help of masks or regions of interest generated by unsupervised segmentation and detection modules, object-level feature alignment further benefits the localization and intra-image contrast (Van Gansbeke et al., 2021; Hénaff et al., 2021; Huang et al., 2022a; Roh et al., 2021; Xie et al., 2021a; Wei et al., 2021). Creating synthetic views by copying regions from other images like mixup augmentation (Zhang et al., 2017) could also obtain prior foreground masks (Wang et al., 2022; Yang et al., 2021). However, most of the above feature alignments are combined with the original image-level loss. How to balance multi-level supervision and avoid over-weight remains further exploration.

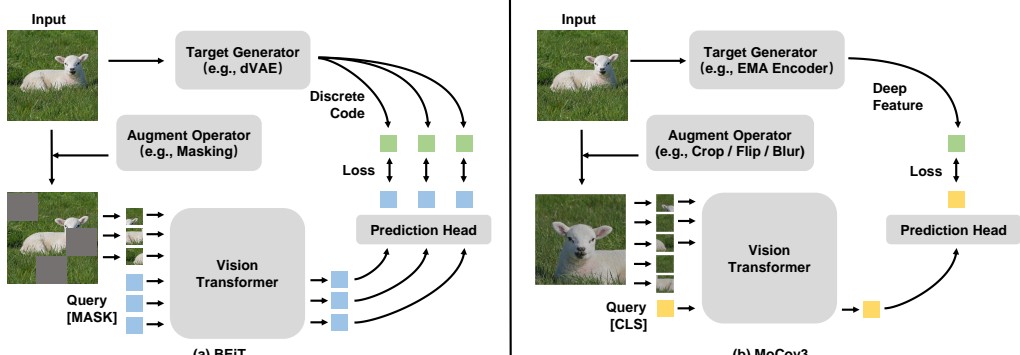

Figure 2: A unified view of the masked image modeling (*i.e.,* BEiT) and contrastive learning (*i.e.,* MoCov3) paradigm. The augment operator transforms input image into another view while preserving semantic information. The target generator produces ground truth. The vision transformer and prediction head are trained modules.

## 3 METHOD

### 3.1 A UNIFIED VIEW OF SELF-SUPERVISED LEARNING FRAMEWORK

In this section, we present a unified view of the self-supervised learning (SSL) framework, aiming to harmonize the merits of masked image modeling (MIM) and contrastive learning (CL). It is worth noting that self-supervised algorithms are a large family containing many types of pretext tasks. The framework presented here only focuses on algorithms used for training vision transformers. Besides, we include methods like BYOL (Grill et al., 2020) into "CL" category as well since they still follow the similar rule that aligning global features.

The MIM framework generally consists of four principal components: an augment operator, an encoder, a prediction head, and a target generator. As shown in Fig. 2 (a), an input image $x$ is augmented by random masking to remove a certain proportion of image patches. Define the index set of all image patches as $I = \{1, \ldots, N_{[\text{IMG}]}\}$, the index set of the remaining image patches and discarded image patches as $R_{[\text{IMG}]} \subseteq I$ and $R_{[\text{MASK}]} = I - R_{[\text{IMG}]}$, respectively. The retained patches are projected into a series of patch embeddings $\{x_i\}$, whereas the discarded ones are substituted with the equivalent number of repeated learnable query tokens, known as [MASK] tokens in MIM. After applying positional embeddings to the sequence according to their position index from $R_{[\text{IMG}]}$ and $R_{[\text{MASK}]}$, the whole sequence is then processed by the encoder and the prediction head. Meanwhile, the target generator takes the complete original image as input and generates the dense target $t_i$, *i.e.,* the supervision signal. Finally, the reconstructed [MASK] tokens, restoring the semantic information of the corresponding missing patches, are supervised by the generated target with the designated similarity measure $\mathcal{L}$. The objective function is delineated as follows:

$$\min \mathop{\mathbb{E}}_{x \in \mathcal{D}} \mathop{\mathbb{E}}_{i \in R_{[\text{MASK}]}} \mathcal{L}(z_i, t_i), \tag{1}$$

where $\mathcal{D}$ is the training corpus and $z$ is the output feature of the prediction head. The choice of the $\mathcal{L}$ depends on the target used. For example, $\ell_1$ is used when using pixel (Xie et al., 2022) or HOG (Wei et al., 2022a) as targets, and cross-entropy is used when the target is produced by discrete tokenizers such as dVAE (Bao et al., 2022) or VQGAN (Dong et al., 2023).

Further, we find that the CL methods, particularly those employing self-distillation manner, share a similar framework as MIM methods as illustrated in Fig. 2(b). For example, MoCo (Chen* et al., 2021), a typical CL approach, employs augmentation techniques such as random resized cropping to generate multiple views of an input image. The query to obtain representation with global semantics is a single learnable embedding, known as [CLS] token in vision transformer. The target generator is a dual online encoder updated by exponential moving average (EMA). The generated deep feature target is aligned with the output query token by contrastive loss(Oord et al., 2018).

Both methodologies adhere to a common paradigm of extracting and aligning features from augmented views of the same image. The difference lies in the access of features and type of target.

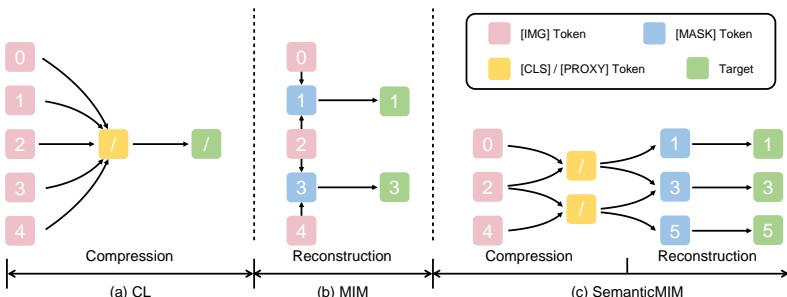

Figure 3: Information propagation of the contrastive learning, masked image modeling, and our proposed SemanticMIM. Numbers indicates position ids and the slash means position-irrelevant. The compression structures present in CL methods endow the model with a better ability to capture semantics. Inspired by this, we introduce similar compression structures into MIM, aiming to enhance the semantic awareness on top of the original positional awareness of MIM.

More specifically, MIM uses several queries with positional prior to focus on local neighbor pixels with *positional awareness* and uses *dense* target for supervision, whereas CL only uses a single query without prior to obtain features with *global semantics* and uses *image-level* target. Previous works have demonstrated that the four components of the framework are replaceable between MIM and CL methods. For instance, the dual EMA encoder in CL could also serve as the target generator in MIM (Tao et al., 2023; Chen et al., 2022; Dong et al., 2022; Zhou et al., 2021), and the masking operation in MIM can join the augmentation series in CL to generate more challenging views (Assran et al., 2022; Huang et al., 2023; Shen et al., 2023). This cross-utilization highlights the flexibility and shared foundational principles of both methodologies.

### 3.2 DISCUSSION ON PROPERTIES OF SELF-SUPERVISED LEARNING FRAMEWORK

Self-supervised learning aims to train models that exhibit robust generalization across various downstream tasks. To give a more specific definition, an ideal pre-trained model is expected to be capable of encoding features of promising consistency and completeness. Consistency ensures that queries on identical objects elicit similar responses. Completeness guarantees that arbitrary objects within an image, including backgrounds, should be encoded into features of the corresponding positions.

CL methods exhibit promising consistency but poor completeness. They could adeptly capture the salient objects. However, this focus comes at the expense of completeness, as they tend to neglect the details of the background. In contrast, MIM achieves high completeness by capturing detailed representations across the entire image but struggles with consistency. More specifically, the redundancy in features of MIM means they are more likely to exhibit response by neighbor patches rather than those of similar semantics. The low consistency underlines MIM's challenge in capturing global semantic representations. From Sec. 3.1, we have pointed out that the difference between MIM and CL may stem from their distinct target and query. This section delves into how *target* and *query* contribute to consistency and completeness.

**Dense target encourages completeness but reduces consistency.** Regardless of the specific target generator employed, MIM inherently produces dense targets characterized by significant redundancy. More specifically, This redundancy implies that the targets for neighboring areas bear a strong similarity to each other. This phenomenon essentially degenerates MIM into a variant of an autoencoder, where the model (*i.e.,* [MASK] query) tends to replicate its neighbors. We identify it as a "learning shortcut" inherent to MIM, leading to a limited receptive field that inadvertently encourages a model to mimic the properties of adjacent areas rather than understanding context.

To mitigate this issue, most MIM framework adopts a high mask ratio to reduce the probability that adjacent patches exist and compel the model to extend its focus to the broader context rather than local neighbors. Advanced masking strategy (*e.g.,* SimMIM (Xie et al., 2022)) is also designed to exclude adjacent patches by using the mask unit of larger size. But as shown in the Fig. 1, not all

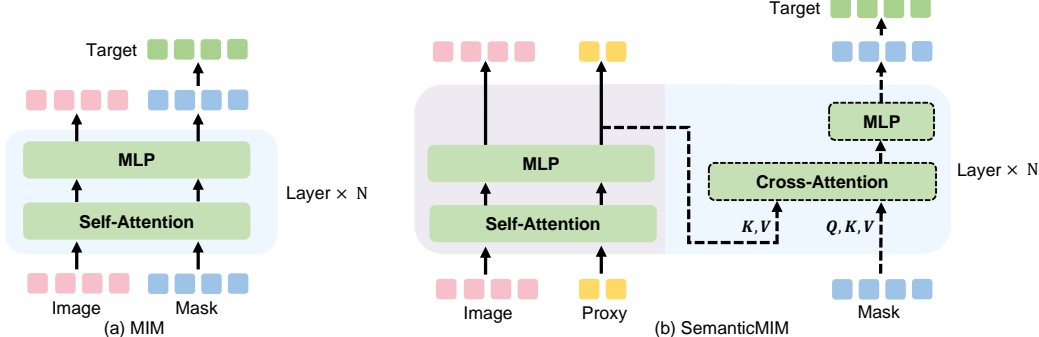

Figure 4: Comparison of the architecture. MIM only focuses on Reconstruction. In SemanticMIM, since the number of `[PROXY]` is much smaller than that of `[IMG]`, information is compressed first (left) and then transmitted to `[MASK]` to complete the reconstruction (right). This design introduces compression while remaining compatible with the original MIM framework.

features belonging to a specific object have high similarity but only those adjacent in spatial do, indicating that the MIM model still tends to focus on local areas and struggles with low consistency.

**Global target encourages consistency but reduces completeness.** CL methods utilize a single global feature as the target, generated either by a sophisticatedly trained model or an online EMA updated encoder. This global feature target typically encapsulates the essence of the foreground at the detriment of background details. Meanwhile, compared to the dense supervision of MIM, a single target feature in CL is high-level and more conceptual, raising properties that explicitly contain the foreground layout of high consistency.

**Query in CL acts compression and queries in MIM find neighbors.** We provide analysis from the perspective of information propagation. In CL, the `[CLS]` token serves as the query, embodying a mechanism that captures global semantics and generates abstract features. Its capability is empowered by the implicit compression during forwarding, as shown in Fig. 3(a). `[CLS]` token is tasked with compressing information from all relevant image patches, aligning itself with targets that encapsulate global context. This compression phase helps the model retain the most essential information and reduce feature redundancy. This single query token `[CLS]`, however, provides limited capacity and over compress information.

Conversely, MIM employs learnable semantic-free embeddings, the `[MASK]` tokens as queries. Distinct from the query in CL, the `[MASK]` tokens are applied with position embeddings as prior to indicate target reconstruction areas. However, the prior inadvertently encourages convenience to the "learning shortcut" mentioned in the last section, making it effortless for the model to locate the neighboring image patches of the masked patches and reduce the necessity of leveraging broader contexts. As depicted in Fig. 3(b), when the queries only propagate information with a limited number of neighboring image patches, the pre-trained model of MIM performs poor feature consistency and notable redundancy without the help of compression.

### 3.3 MASKED IMAGE MODELING WITH PROXY ARCHITECTURE

Based on the provided analysis, we propose a neat framework SemanticMIM drawing inspiration from the compression of CL and applying it to solve the inherent limitations of MIM, as shown in Fig. 3(c). To mitigate the issue of easily locating neighboring image patches of the masked patches due to the positional prior, we disrupt the direct information propagation between `[IMG]` tokens and the `[MASK]` tokens. Instead, we leverage extra tokens with no positional prior in between as a proxy, naming it `[PROXY]` token. Second, since the `[PROXY]` tokens and `[MASK]` tokens are both queries with no semantic information, the information is forced to spread from `[IMG]` tokens to `[PROXY]` tokens and then from `[PROXY]` tokens to `[MASK]` tokens. The `[PROXY]` token plays a role in the information bottleneck and thus the original pretext task is divided into two distinct stages: *compression* and *reconstruction*. We can calibrate the extent of compression by adjusting the number of `[PROXY]` tokens used.

The implementation is shown in Fig. 4. In the original MIM framework, [IMG] and [MASK] tokens are processed as a whole sequence. Suppose the hidden state of the [IMG] and [MASK] tokens of layer $i$ as $\boldsymbol{h}^i_{[\texttt{IMG}]}$ and $\boldsymbol{h}^i_{[\texttt{MASK}]}$, respectively. The forward process in each transformer layer is defined as follows:

$$\boldsymbol{h}^{i+1}_{[\texttt{IMG}]}, \boldsymbol{h}^{i+1}_{[\texttt{MASK}]} = \text{MLP}(\text{SelfAttn}([\boldsymbol{h}^i_{[\texttt{IMG}]}, \boldsymbol{h}^i_{[\texttt{MASK}]}])). \tag{2}$$

The key idea of SemanticMIM is a specific mechanism of information propagation constraint. For the three types of tokens, our goal is to architecturally segregate [IMG] and [MASK], rendering them mutually exclusive in visibility, while simultaneously ensuring both are accessible to [PROXY] tokens. This is achieved through a modification of the transformer block, as illustrated in Fig. 4(b), which incorporates dual cascaded attention and MLP modules, mirroring settings across each layer. The self-attention and subsequent MLP only process [IMG] and [PROXY] tokens, responsible for the compression task, gathering semantic information from image patches and compressing it into [PROXY] tokens. This forward can be formulated as Eq. (3), where $\boldsymbol{h}^i_{[\texttt{PROXY}]}$ is the hidden state of the [PROXY] token at layer $i$. The extra cross-attention and the following MLP finish the reconstruction task. In particular, the sequence formed by concatenating [PROXY] and [MASK] tokens is utilized as key and value, while only the [MASK] token serves as query input. We formulate this process as Eq. (4).

$$\boldsymbol{h}^{i+1}_{[\texttt{IMG}]}, \boldsymbol{h}^{i+1}_{[\texttt{PROXY}]} = \text{MLP}(\text{SelfAttn}([\boldsymbol{h}^i_{[\texttt{IMG}]}, \boldsymbol{h}^i_{[\texttt{PROXY}]}])) \tag{3}$$

$$\boldsymbol{h}^{i+1}_{[\texttt{MASK}]} = \text{MLP}(\text{CrossAttn}(\boldsymbol{h}^i_{[\texttt{MASK}]}, [\boldsymbol{h}^{i+1}_{[\texttt{PROXY}]}, \boldsymbol{h}^i_{[\texttt{MASK}]}])) \tag{4}$$

With this design, the compression and reconstruction task is fully disentangled and executed by independent modules. Such disentanglement makes the reconstruction modules serve as a dedicated plugin for the pre-training stage and can be discarded later. Besides, calculating attentions separately allows SemanticMIM to have the same or even lower computational cost than vanilla MIM. Detailed analysis is provided in Appendix C. Further, the encoder only performs the compression task in our framework, avoiding wasting capacity on the reconstruction task as in the original MIM framework (Liu et al., 2023). Our design better meets the requirements of the downstream tasks for discriminative visual representations with consistency and completeness defined in Sec. 3.2.

## 4 EXPERIMENTS

### 4.1 PRE-TRAINING SETTING

As our proposed method only modifies the way that information passed in the encoder, it is parallel to any MIM framework. The additional [PROXY] tokens are indeed learnable embeddings just like the [CLS] token, in the implementation, we initialize multiple [CLS] tokens and use them as the [PROXY] tokens. Notably, the [PROXY] tokens only acts as springboards and are not supervised by any signals directly. So we compute loss only on [MASK] tokens as in the original.

To illustrate the generality, we choose two representative baselines BEiT (Bao et al., 2022) and MaskFeat (Wei et al., 2022a), which utilize high-level and low-level targets respectively. For all experiments, we use ViT-Base (Dosovitskiy et al., 2021) with patch size 16 as the encoder backbone and pretrain it on ImageNet-1K (Deng et al., 2009) dataset over 300 epochs at $224^2$ resolution. Both baselines adopt 40% mask ratio and one [CLS] token. When applying our methods, we set the mask ratio to 60% and the number of [PROXY] tokens to 8. The model used in ablation study and visualization is based on BEiT unless specified otherwise. Further details on our pre-training are provided in the Appendix E.

### 4.2 EVALUATION SETTINGS

To quantitatively validate the effectiveness of our methods, we conduct experiments on classification and semantic segmentation tasks with both linear probing and end-to-end fine-tuning. Further details are described in the Appendix E.

Table 1: **Performance comparison with baselines.** We report top-1 accuracy on ImageNet-1K, mIoU on ADE20K, and mIoU on PascalVOC. Linear and FT stand for linear probing and fine-tuning, respectively. All results are produced by ourselves.

| Datasets | ImageNet-1K | | | | PascalVOC | ADE20K |
|---|---|---|---|---|---|---|
| Protocol | Linear | | FT | | Linear | FT |
| Feature | CLS | Patch | CLS | Patch | Featmap | Featmap |
| BEiT | 31.5 | 38.7 | 81.9 | 82.2 | 23.8 | 40.2 |
| BEiT + Ours | 49.2(+17.7) | 48.2(+9.5) | 83.0(+1.1) | 82.9(+0.7) | 43.1(+19.3) | 44.1(+3.9) |
| MaskFeat | 23.4 | 33.5 | 82.7 | 83.0 | 37.8 | 42.6 |
| MaskFeat + Ours | 52.0(+28.6) | 59.7(+26.2) | 83.7(+1.0) | 83.6(+0.6) | 49.5(+11.7) | 45.7(+3.1) |

For the classification task, we train a supervised linear classifier on the ImageNet-1K training set for 100 epochs and report top-1 accuracy on the ImageNet-1K validation set following the settings in (Bao et al., 2022). The classifier is integrated at the final layer under the fine-tuning protocol and at the $7^{th}$ layer to harness a generalizable feature representation under the linear probing protocol. Additionally, we provide results of feeding the classifier with [CLS] tokens (named CLS in the table) and with average pooling features of output [IMG] tokens (named Patch).

For the semantic segmentation task, we report mIoU on ADE20K (Zhou et al., 2017) benchmark with 150 semantic categories for end-to-end fine-tuning and PascalVOC (Everingham et al., 2010) benchmark with 21 semantic categories for linear probing. More specifically, on ADE20K, we use UperNet (Xiao et al., 2018) as the decoder and fine-tuning for 160k steps at $640^2$ resolution following (Bao et al., 2022). On PascalVOC, we train a $1 \times 1$ conv layer on top of the frozen 6-th layer feature at $448^2$ resolution for 25 epochs following (Ziegler & Asano, 2022).

## 4.3 MAIN RESULTS

We validate our proposed SemanticMIM by incorporating it to BEiT and MaskFeat in Tab. 1. On classification, SemanticMIM enhances accuracy by 10% for BEiT and 25% for MaskFeat during linear probing, and around 1% under fine-tuning. On segmentation, our method outperforms BEiT by 19.3% and MaskFeat by 11.7% under linear probing and around 3% under fine-tuning.

Three findings emerge from the results. First, SemanticMIM notably enhances performance under linear probing, a protocol that directly assesses the quality of visual representations, indicating that SemanticMIM learns more linearly separable and discriminative features. Second, our method shows a more pronounced improvement on MaskFeat compared to BEiT. This discrepancy can be attributed to MaskFeat's use of low-level HOG targets, which possess less semantic information and greater redundancy, leading to features encoding with more details but poor consistency. Compression introduced in our framework is particularly effective for this scenario, yielding substantial performance enhancements. Thirdly, with our method, the [CLS] tokens become more adept at extracting global information, since they serve as the proxy to effectively gather information from context and learn compressed semantic features without supervision.

Note that the baseline performance of our reproduced BEiT and MaskFeat is lower than reported in the original paper and it is mainly due to the different settings. BEiT and MaskFeat use block-wise masking and undergo training for 800 and 1600 epochs, respectively. Our study employs random patch masking and limits training to 300 epochs for simplification purposes.

## 4.4 ABLATION STUDY ON NUMBER OF [PROXY] TOKEN

Fig. 5 shows the impact of the number of [PROXY] tokens in our method. We train ViT-Base for 300 epochs and then fine-tune for 100 epochs. ImageNet-1K validation accuracy under fine-tuning protocol is reported. The resolution of the input image is 224x224 and patch size is 16, so the number of [IMG] tokens is 196. As the number of [PROXY] tokens increases, the extent of compression decreases, more information is transferred to the [MASK] token for reconstruction and thus the task difficulty is reduced. The optimal performance is achieved with 8 [PROXY] tokens.

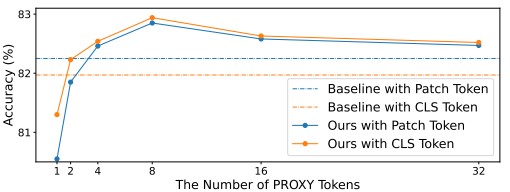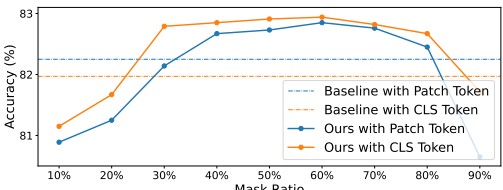

Figure 5: **Ablation on [PROXY] tokens**.    Figure 6: **Ablation on mask ratio**.

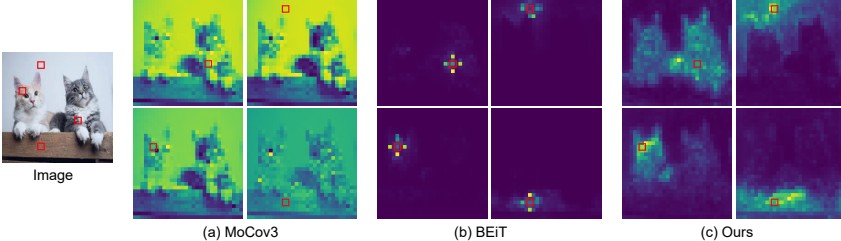

Figure 7: **Attention maps queried by distinct patches across different methods.** The query patches to produce attention maps are marked with red boxes.

SemanticMIM with only 2 [PROXY] tokens achieves competitive performance to the baseline, indicating high redundancy of the original image context. With more [PROXY] tokens, the effect of compression gradually diminishes. Considering the extreme case of using as many [PROXY] tokens as [MASK] tokens or even more, our method degenerates to the conventional MIM except for an extra information exchange between [PROXY] and [MASK] token. Hence, the performance of our methods gradually approaches the baseline as the number of [PROXY] tokens increases.

### 4.5 ABLATION STUDY ON MASK RATIO

Fig. 6 shows the effect of the mask ratio under the same training setting as in Sec. 4.4. A low mask ratio leads to overly rich context information rendering the pretext task insufficiently challenging, and vice versa for a high mask ratio. The optimal ratio of our method is around 60%. Previous works whose encoder processes only visible patches use a higher ratio like 75% in MAE (He et al., 2022) and those processing the whole sequence including [MASK] tokens use a lower mask ratio like 40% in BEiT (Bao et al., 2022). Our architecture is similar to MAE, in which the encoder does not process [MASK] tokens. The information bottleneck brought by [PROXY] tokens increases the task difficulty compared to the original MIM framework, thus lowering the optimal mask ratio.

## 5 VISUALIZATION

In this section, we provide a qualitative analysis by visualizing the attention response of the pre-trained models. We compare MoCov3 (Chen* et al., 2021), BEiT (Bao et al., 2022), and our method based on BEiT to explore the properties of CL, MIM, and the proposed SemanticMIM. More examples are provided in the Appendix D.

**SemanticMIM satisfies both completeness and consistency.** As shown in Fig. 7, we visualize the attention map with [IMG] as queries. MoCov3 displays a lack of positional sensitivity and generates homogeneous attention maps that distinguish foreground and background regardless of which image patch to query. BEiT suffers from local receptive fields and only neighbor image patches respond to the query. SemanticMIM integrates the advantages of both, being position-aware and semantic-aware. All [IMG] tokens belonging to the same object as the given patch respond to the query, which illustrates the remarkable consistency. Besides, all objects in the foreground and background have correct and distinct responses, showcasing strong completeness. Moreover, a notable observation is that SemanticMIM assigns similar features to the two cats with different appearances, underscoring its capability of semantic perception.

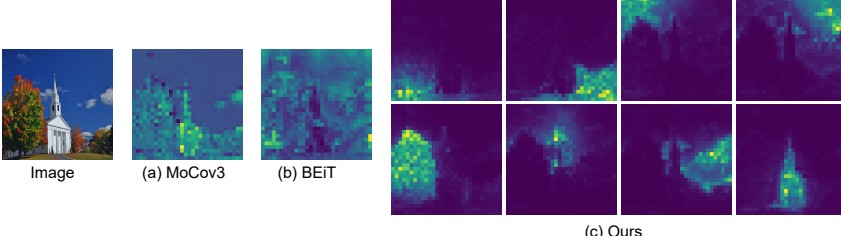

Figure 8: **Attention maps queried by `[CLS]`/`[PROXY]` token across different methods.**

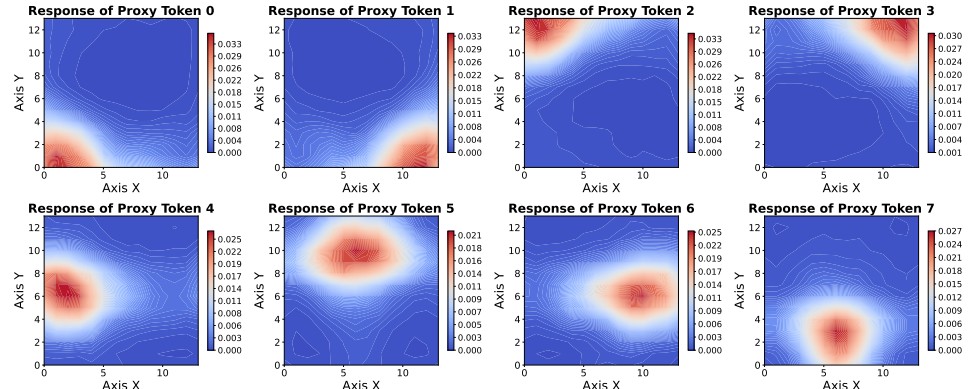

Figure 9: **Heatmap of the focused area of distinct `[PROXY]` tokens.** We average the attention map of 500 images from ImageNet to display this pattern.

**Deciphering the intrinsic mechanism of SemanticMIM.** Fig. 8 unveils the attention response of the `[PROXY]` token. Supervised by the global deep feature target, the `[CLS]` token of MoCov3 focuses on the foreground, including the trees and the tower. In contrast, BEiT lacks explicit `[CLS]` token supervision and the disorderly attention response illustrates that it struggles to gather semantic information. For SemanticMIM, different `[PROXY]` tokens pay attention to objects of different regions and most of the patches respond to the query in each attention map belonging to the corresponding semantic category.

Delving deeper, we calculate the average attention map of each `[PROXY]` token over 500 images from ImageNet, and the result is shown in Fig. 9. It is observed that each `[PROXY]` token focuses on almost exclusive regions. Since an image patch of arbitrary position may be selected for reconstruction, the `[PROXY]` tokens, the only information provider for the mask tokens, are forced to encode information of all regions. The most efficient encoding method is that each `[PROXY]` tokens store information of distinct areas non-overlappingly. Hence, the `[PROXY]` tokens in SemanticMIM tend to region-level object queries with position prior, gathering semantic information from regions of interest, which facilitates the reconstruction by encouraging `[MASK]` token to explore region-level context.

## 6 CONCLUSION

In this paper, we present SemanticMIM to integrate the merit of contrastive learning into masked image modeling. We first abstract the essence of CL and MIM to compression and reconstruction through comprehensive analysis. With this hypothesis, SemanticMIM naturally leverages a proxy architecture to first compress all information of `[IMG]` token into `[PROXY]` token, and reconstruct `[MASK]` token conditioned on these `[PROXY]` token. As a result, SemanticMIM adeptly models global semantics akin to contrastive learning, while preserving the spatial awareness intrinsic to masked image modeling, leading to a general self-supervised visual representation. Further, extensive qualitative and quantitative experiments validate the effectiveness of SemanticMIM.

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

## A ANALYSIS ON ATTENTION DISTANCE

Following (Xie et al., 2023b), we analyze the average attention distance across models pre-trained by three distinct methods, *i.e.* MIM (BEiT), CL (MoCov3), and our proposed SemanticMIM, as shown in Fig. 13. The attention distance is computed by averaging the distance between the query patch and all other patches, weighted by the attention weights (Dosovitskiy et al., 2021). It is analogous to the receptive field where higher value refers to a broader context dependency.

We observe that CL pre-trained models tend to focus on the local context at lower layers, transitioning to more global context at higher layers, while MIM pre-trained models display an opposite trend. SemanticMIM, although grounded in MIM's training architecture, exhibits a pattern akin to CL, suggesting that data compression plays a pivotal role in managing context dependency. Meanwhile, SemanticMIM retains MIM's characteristic of diverse head behaviors across all layers. Finally, it is observed that the average attention distance of SemanticMIM is higher than MIM and lower than CL. We argue that MIM might overly concentrate on neighboring patches, while in CL the entire foreground containing multiple objects responds to the query. SemanticMIM can distinguish objects of different semantic categories, leading to a balanced attention distance.

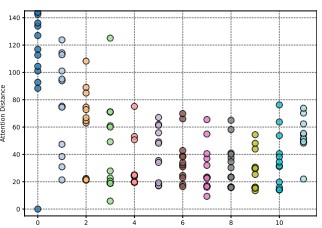 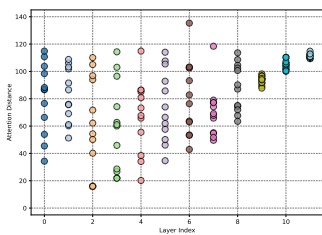 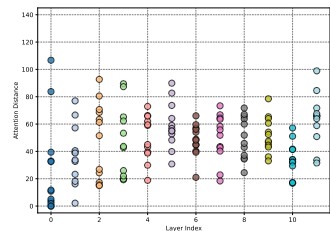

Figure 10: BEiT.  Figure 11: MoCov3.  Figure 12: SemanticMIM.

Figure 13: **Comparison of averaged attention distance across different types of self-supervised methods.** The y-axis refers to the averaged attention distance, while x-axis represents the layer index. Each data point for a given layer index represents a specific attention head. The baseline of SemanticMIM is BEiT. We reproduce BEiT by ourselves and use the released weights from the official MoCov3.

## B VISUALIZATION OF ATTENTIONS GROWING ALONG LAYERS

As shown in Fig. 14, we visualize the attention map across various layers. The pretrained model is ViT-B with 16 [PROXY] tokens under MaskFeat framework. It indicates that attention in the shallow layers predominantly focuses on the local neighbors of the query patch. With the depth increases, the response area in the attention map gradually broadens, indicating that the model progressively explores the context with further spatial distance. Finally, the attention map converges to the semantic layout of the corresponding object. Previous work (Xie et al., 2023b) has shown that supervised pre-trained and CL pre-trained models tend to exhibit a shift from local to global focus across layers but MIM pre-trained model brings locality inductive bias. SemanticMIM follows the framework of MIM and behaves like CL and supervised pre-training, indicating that compression is crucial to the global receptive field.

## C ANALYSIS ON COMPUTATIONAL COST AND PARAMETERS

The introduction of SemanticMIM brings a slight change in computational cost. Suppose the shape of the input image tensor is $[B, L, D]$. A standard ViT block consumes:

$$Cost_{Attn} = 4BLD^2 + 2BL^2D$$
$$Cost_{MLP} = 8BLD^2$$
$$Cost_{Total} = 12BLD^2 + 2BL^2D$$

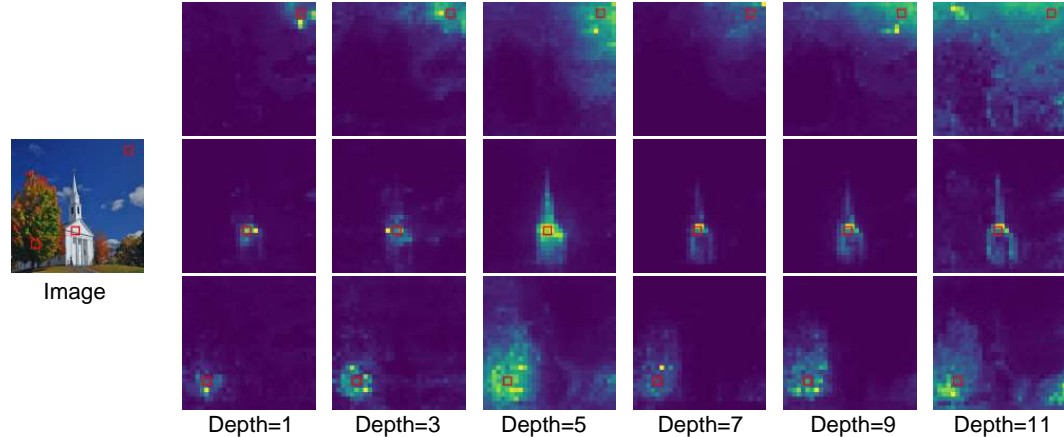

Figure 14: **Attention maps of SemanticMIM across depths.** The patches used as queries are marked with red boxes and the depth refers to the layer index.

Specifically, we suppose that $L_0, L_1, L_2$ refer to the number of [IMG], [PROXY] and [MASK] tokens, and $L = L_0 + L_1 + L_2$ where $L_1 = 0$ for vanilla MIM. In SemanticMIM, the cost becomes:

$$Cost_{Attn} = 4B(L_0 + L_1)D^2 + 2B(L_0 + L_1)^2 D$$
$$Cost_{MLP} = 8B(L_0 + L_1)D^2$$

As for the cross-attention module, its query has the shape $[B, L_2, D]$ and its key and value have the shape $[B, L_1 + L_2, D]$, thus consumes

$$Cost_{Cross} = 3BL_2D^2 + BL_1D^2 + 2BL_1L_2D + 2BL_2^2D$$

In total, a semanticMIM block consumes

$$Cost_{Total} = 12B(L_0 + L_1)D^2 + 2B(L_0 + L_1)^2 D$$
$$+ 11BL_2D^2 + BL_1D^2 + 2BL_1L_2D + 2BL_2^2D$$

During training, take our ViT-B setting as an example where $D = 768, L_1 = 8, L_0 = 78, L_2 = 118$, SemanticMIM achieves a **2.8% reduction** in FLOPs compared to vanilla MIM. The computational cost becomes equivalent when $L_1 = 11$. During inference, where $L_0 = 196, L_2 = 0$, using 8 proxy tokens ($L_1 = 8$) leads to only 3.7% increase in FLOPs, which is accompanied by a considerable performance gain. As for parameters, since the cross-attention modules are discarded after training, the only difference between semanticMIM and vanilla MIM models are a few proxy tokens, resulting in a negligible increase compared to the whole model.

## D VISUALIZATION ON MORE SCENARIOS

In this section, we present more attention visualization results in Figs. 15 and 16. We pre-train a ViT-Base model with our proposed SemanticMIM framework based on BEiT on ImageNet-1K for 300 epochs. We evaluate SemanticMIM under both simple and complex scenarios without selective cherry-pick.

## E DETAILED RECIPES

We provide the detailed recipe of pre-training in Tab. 2 and all four evaluation experiments in Tab. 3, Tab. 4, Tab. 5, and Tab. 6.

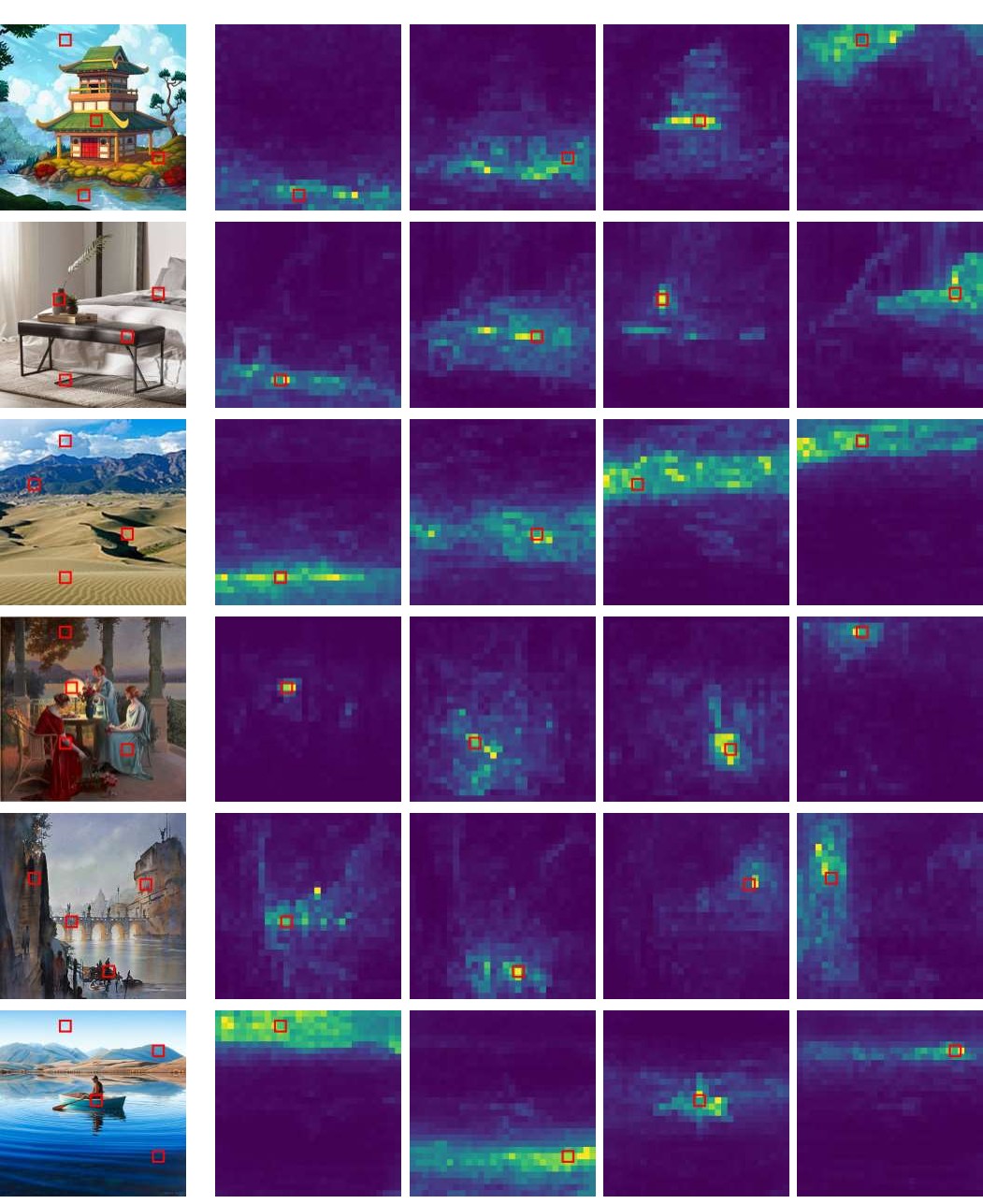

Figure 15: **Attention maps of complex scenarios.** We select several different styled images from LAION (Schuhmann et al., 2022), containing multiple objects as inputs. The queried patches are marked with red boxes.

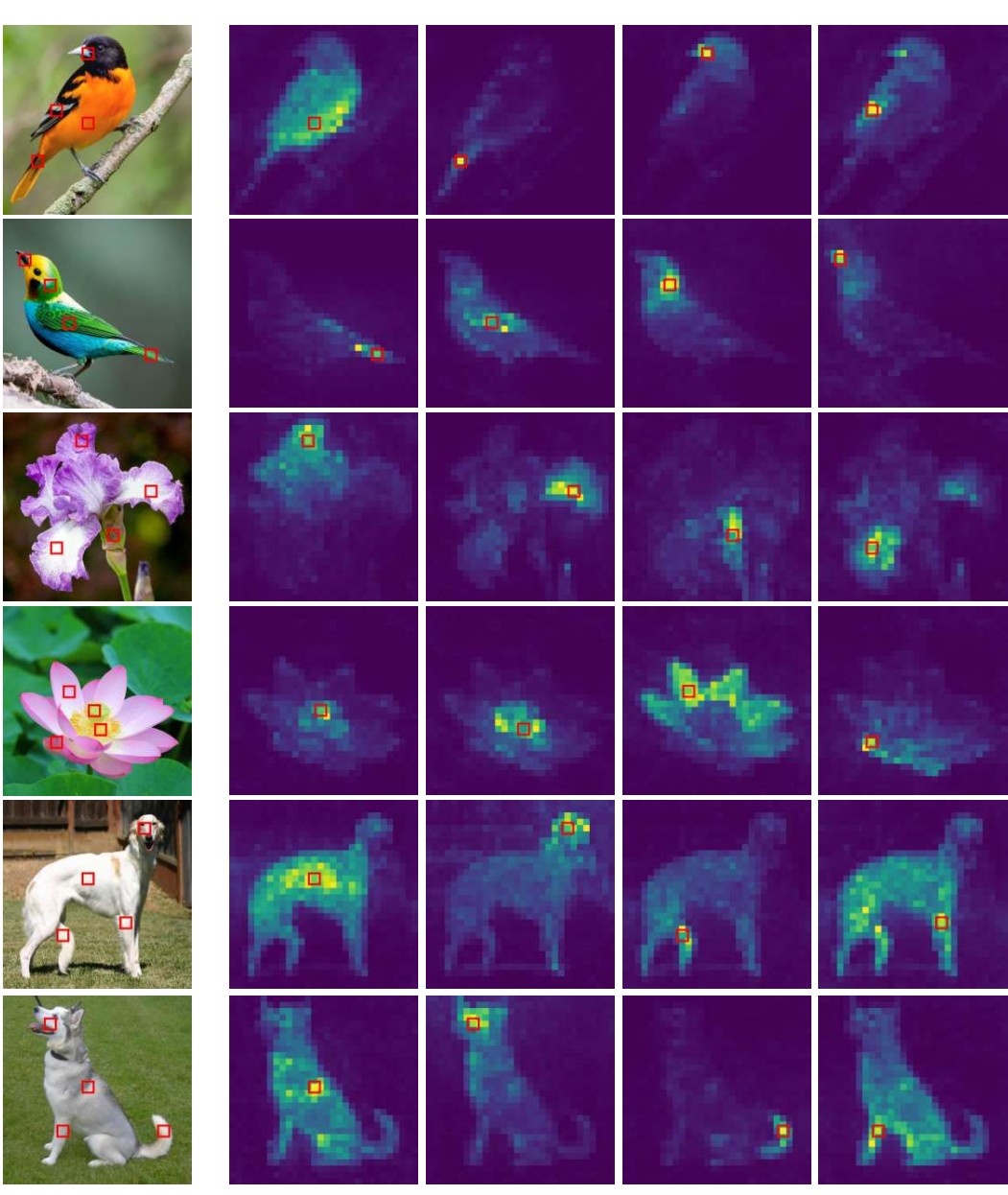

Figure 16: **Attention maps of images containing a single object.** We select several images commonly used in fine-grained classification tasks. The queried patches are marked with red boxes.

Table 2: Hyperparameters for pre-training BEiT and MaskFeat on ImageNet-1K. When applying our proposed method, we use exactly the same recipe.

| Hyperparameters | BEiT | MaskFeat |
|---|---|---|
| Training epochs | 300 | |
| Batch size | 2048 | |
| Adam $\epsilon$ | 1e-8 | |
| Adam $\beta$ | (0.9, 0.999) | |
| Peak learning rate | 1.5e-3 | 1.6e-3 |
| Minimal learning rate | 1e-5 | 1e-6 |
| Learning rate schedule | Cosine | |
| Warmup epochs | 10 | 30 |
| Gradient clipping | 3.0 | 0.02 |
| Stoch. depth | 0.1 | |
| Weight decay | 0.05 | |
| Crop Ratio | (0.08, 1.0) | (0.5, 1.0) |
| Flip Prob | 0.5 | |
| Color jitter | 0.4 | ✗ |

Table 3: Hyperparameters for fine-tuning pre-trained model on ImageNet-1K.

| Hyperparameters | BEiT & MaskFeat |
|---|---|
| Epochs | 100 |
| Batch size | 1024 |
| Adam $\epsilon$ | 1e-8 |
| Adam $\beta$ | (0.9, 0.999) |
| Peak learning rate | 4e-3 |
| Minimal learning rate | 0 |
| Learning rate schedule | Cosine |
| Warmup epochs | 0 |
| Gradient clipping | ✗ |
| Stoch. depth | 0.1 |
| Weight decay | 1e-4 |
| Crop Ratio | (0.08, 1.0) |
| Flip Prob. | 0.5 |

Table 4: Hyperparameters for fine-tuning pre-trained model with UperNet on ADE20K. BEiT and MaskFeat use the same recipe.

| Hyperparameters | BEiT & MaskFeat |
|---|---|
| Fine-tuning Steps | 160k |
| Batch size | 16 |
| Adam $\epsilon$ | 1e-8 |
| Adam $\beta$ | (0.9, 0.999) |
| Peak learning rate | 3e-5 |
| Minimal learning rate | 0 |
| Learning rate schedule | Linear |
| Warmup steps | 1500 |
| Gradient clipping | ✗ |
| Stoch. depth | 0.1 |
| Weight decay | 0.05 |
| Input resolution | $640 \times 640$ |
| Multi-scale Inference | ✗ |

Table 5: Hyperparameters for training the classifier while freezing the pre-trained model following linear probing protocol on ImageNet-1K. BEiT and MaskFeat use the same recipe.

| Hyperparameters | BEiT & MaskFeat |
|---|---|
| Epochs | 100 |
| Batch size | 1024 |
| Adam $\epsilon$ | 1e-8 |
| Adam $\beta$ | (0.9, 0.999) |
| Peak learning rate | 4e-3 |
| Minimal learning rate | 0 |
| Learning rate schedule | Cosine |
| Warmup epochs | 0 |
| Gradient clipping | ✗ |
| Weight decay | 1e-4 |
| Crop Ratio | (0.08, 1.0) |
| Flip Prob. | 0.5 |

Table 6: Hyperparameters for training the segment head while freezing the pre-trained model following linear probing protocol on PascalVOC. BEiT and MaskFeat use the same recipe. For faster training, we interpolate the groundtruth and output to training resolution and use normal eval resolution during evaluation.

| Hyperparameters | BEiT & MaskFeat |
|---|---|
| Epochs | 25 |
| Batch size | 120 |
| Optimizer | SGD |
| Learning rate | 0.01 |
| Learning rate schedule | Step |
| Warmup epochs | 0 |
| Gradient clipping | ✗ |
| Weight decay | ✗ |
| Input resolution | $448 \times 448$ |
| Training resolution | $100 \times 100$ |
| Eval resolution | $448 \times 448$ |
| Multi-scale Inference | ✗ |

