# OpenReview forum: "SemanticMIM: Marring Masked Image Modeling with Semantics Compression for General Visual Representation"
_ICLR.cc/2025/Conference — Submitted to ICLR 2025_

### Official Review · Reviewer_WeyN · 2024-10-19

**Soundness:** 2
**Presentation:** 2
**Contribution:** 3
**Rating:** 5
**Confidence:** 3

**Summary:**

This paper proposes SemanticMIM, a framework that integrates the strengths of masked image modeling (MIM) and contrastive learning (CL) to achieve general visual representation learning. Specifically, SemanticMIM leverages a proxy architecture to compress information from image tokens into proxy tokens and then reconstructs masked tokens, thereby combining the semantic awareness of CL with the spatial sensitivity of MIM. As the approach only modifies the encoder architecture, it can be seamlessly integrated into any MIM framework. Extensive experiments further validate the effectiveness of the model and offer interpretability through attention visualization.

**Strengths:**

1. The analysis of the fundamental principles underlying CL and MIM is insightful and thought-provoking.

2. The proposed SemanticMIM is well-motivated, which leverages the strengths of both CL and MIM.

3. SemanticMIM improves the performance of two MIM methods when applied using the training scheme introduced in this paper.

**Weaknesses:**

1. Some important details about the training process are missing. For instance, are the [PROXY] tokens supervised with specific targets, similar to the supervision used in CL? Are the [MASK] tokens trained using the same loss function employed in typical MIM frameworks? Clarifying these aspects would enhance the reproducibility and understanding of the proposed approach.

2. The classification of SSL methods into only CL and MIM may not be entirely accurate. Earlier works also introduced a variety of pretext tasks, such as image colorization and rotation prediction, which play an important role in the development of SSL methods. A more comprehensive overview of these earlier approaches can be found in the survey “Self-supervised visual feature learning with deep neural networks: A survey”.

3. The reproduced performance of BEiT and Maskfeat in this paper appears much lower than the original results reported in their respective papers. Could you clarify why the official training schemes were not followed? Additionally, if the official training schemes were applied, how much improvement would SemanticMIM bring to these MIM methods?

4. The citation format in some sections of the text requires correction. For example, “Self-supervised learning (SSL) algorithms Liu et al. (2021); Balestriero et al. (2023) have emerged as …” should be revised to “Self-supervised learning (SSL) algorithms (Liu et al. 2021; Balestriero et al. 2023) have emerged as …” to conform to general academic writing practices.

**Questions:**

From Fig. 14, it appears that different layer depths yield different types of attention maps. In Fig. 1, are all the displayed attention maps derived from the same layer depth? Providing this clarification will help readers better understand the visual interpretability.

---

> ### Author Response · Authors · 2024-11-23
> **Reponse to Reviewer WeyN**
>
> Thank you for your close reading and valuable feedback. Our response to your question is as follows.
>
> > Some important details about the training process are missing. For instance, are the [PROXY] tokens supervised with specific targets, similar to the supervision used in CL? Are the [MASK] tokens trained using the same loss function employed in typical MIM frameworks? Clarifying these aspects would enhance the reproducibility and understanding of the proposed approach.
>
> The [PROXY] tokens are not supervised by any signals and are only responsible for information exchange between image and mask tokens. And yes, the [MASK] tokens are trained using the same loss as in the typical MIM framework. Our method only modifies the model architecture, making it adaptable to any MIM framework. Following your suggestion, we have clarified this in the experiment section and modify Fig.4 accordingly.
>
> > The classification of SSL methods into only CL and MIM may not be entirely accurate. Earlier works also introduced a variety of pretext tasks, such as image colorization and rotation prediction, which play an important role in the development of SSL methods.
>
> Thank you for your supplement. In Section.2, we conducted a rough categorization to facilitate the comparison between two types of methods **used on pretraining vision transformers**: those that complete local signals and those that align global features. We have clarified this, added some background in the manuscript to make it rigorous.
>
> > The reproduced performance of BEiT and MaskFeat in this paper appears much lower than the original results reported in their respective papers. Could you clarify why the official training schemes were not followed? Additionally, if the official training schemes were applied, how much improvement would SemanticMIM bring to these MIM methods?
>
> There are two main reasons. First, for the ease of implementation and reproduction, we use mmpretrain, an open-sourced pretraining toolbox for pytorch, to implement our method. We follow the training recipe they provided, in which the model is pretrained for 300 epochs, instead of 800 epochs in official BEiT and 1600 epochs in official MaskFeat. Second, we remove all the training and finetuning tricks, such as block-wise masking, combining class tokens and patch tokens, stack features from multiple layers, etc. Our goal is to find a clean and simple baseline to prove that our proposed method can effectively capture semantics without relying on other modules. Since all these differences mentioned above are irrelevant to our proposed method, we believe that this does not hinder the correctness of our analysis and the effectiveness of our method. Similar gains can be expected with the official baseline or other MIM methods.
>
> > The citation format in some sections of the text requires correction.
>
> We apologize for our incorrect usage of \cite and have corrected all the citation formats in our manuscript.
>
> > In Fig. 1, are all the displayed attention maps derived from the same layer depth? Providing this clarification will help readers better understand the visual interpretability.
>
> Thanks for your suggestions. The results of MoCo and BEiT are from the 10th layer, while ours are from the 8th layer.  We have clarified this in the Fig.1 caption.

---

> ### Comment · Reviewer_WeyN · 2024-11-24
>
> Thanks for the detailed response, which provides a clearer perspective on the paper. However, I still have one question. Given that the [PROXY] token does not rely on the same form of supervision as contrastive learning and primarily functions as a mechanism for information compression, why is the proposed method described as integrating the merits of contrastive learning into masked image modeling? Would it not be more accurate to frame it as leveraging the benefits of information compression? Alternatively, what specific aspects of the [PROXY] design justify categorizing it as a contrastive learning method? I think clarifying this distinction would help enhance the conceptual understanding of the proposed approach.

---

> > ### Author Response · Authors · 2024-11-25
> > **Response to Reviwer WeyN**
> >
> > Thank you for your timely response.
> >
> > > Given that the [PROXY] token does not rely on the same form of supervision as contrastive learning and primarily functions as a mechanism for information compression, why is the proposed method described as integrating the merits of contrastive learning into masked image modeling?
> >
> > As analyzed in Section 3.1 in our manuscript, CL (Contrastive Learning) and MIM (Masked Image Modeling) actually follow a similar framework when training ViTs. Certain components within the framework can be applied to both CL and MIM methods, hence these are not the key factors of the differing properties observed in models trained by CL and MIM. We believe the main difference lies in the fact that CL involves compression through the [CLS] token, whereas MIM does not. Following this line of thought, we introduced compression via [PROXY] token into MIM, leading to SemanticMIM, which demonstrated a similar capability to capture object semantics as CL, validating our analysis. Additionally, it retained the position-sensitive capability of MIM, which are absent in CL.
> >
> > > Would it not be more accurate to frame it as leveraging the benefits of information compression? Alternatively, what specific aspects of the [PROXY] design justify categorizing it as a contrastive learning method?
> >
> > Indeed, "leveraging the benefits of compression" is an accurate description.
> > The similarity between SemanticMIM and CL is that the PROXY token in SemanticMIM and the CLS token in CL play similar roles, acting as a cache for compression which saves integrated global semantic information. The difference lies in the fact that the [CLS] token in CL directly receives supervision, whereas the [PROXY] tokens passes semantic information to the [MASK] tokens and are supervised indirectly. That's the MIM part in SemanticMIM: learning by inferring missing information from context. Combining the analysis above, we believe describing it as a combination of CL and MIM more closely aligns with our motivations and further conveys our analysis and thoughts on CL and MIM.
> >
> > To avoid confusion, we have emphasized this in the manuscript, that the inspiration SemanticMIM draws from CL-category methods primarily lies in the concept of compression. In Section 3.3, we clarified that "Based on the provided analysis, we propose a neat framework SemanticMIM drawing inspiration from the compression of CL and applying it to solve the inherent limitations of MIM, as shown in Fig. 3(c)."

---

> ### Comment · Reviewer_WeyN · 2024-11-26
>
> Thanks for the continual response. I agree that models trained using CL have the property of information compression. However, I am hesitant to fully agree with the broader claim that models trained to compress information without direct CL-style supervision can automatically be categorized as CL. In my understanding, a defining characteristic of CL is the use of a contrastive loss function during training. While I appreciate the motivation behind the proposed approach, if it is to be described as “a combination of CL and MIM”, I think the [PROXY] tokens would be better to be trained using a CL-style loss, which would better align the method with the proposed terminology. Based on this, I will maintain my current rating.

---

> ### Author Response · Authors · 2024-11-26
> **Response to Reviewer WeyN**
>
> Thanks for your further advices. Your understanding of our approach is accurate. SemanticMIM is not a combination of CL and MIM methods. On the contrary, because it **strictly adheres to the MIM framework**, it is a MIM method that involves compression, which is achieved by [PROXY] serving as a plugin. Therefore, our experiments primarily compare SemanticMIM with other MIM methods to demonstrate the improvements brought by compression.
>
> To avoid confusion caused by broader claim, we have made the following revisions to the manuscript.
>
> 1. We have corrected the relevant statements in the manuscript and emphasized that our method integrates **the advantages or merits** rather than the methods themselves.
> 2. To make this point clearer, we have added the following statement to the introduction section and highlighted it in blue.
> "Inspired by CL, we propose SemanticMIM, a novel paradigm that introducing compression within MIM framework, aiming to harness some of the advantages of CL methods. **It is noteworthy that SemanticMIM is not a combination of CL and MIM in a multi-task manner. Instead, it strictly adheres to the general MIM framework and achieves compression by controlling information exchange.**"

---

### Official Review · Reviewer_Efcx · 2024-10-28

**Soundness:** 3
**Presentation:** 1
**Contribution:** 2
**Rating:** 5
**Confidence:** 5

**Summary:**

This paper proposes a self-supervised learning method to pretrain vision transformers named SemanticMIM. SemanticMIM tries to take the advantage from both masked image modeling and contrastive learning by leveraging a proxy architecture that customizes interaction between image and mask tokens, bridging these two phases to archieve general visual representation with the property of abundant semantic and positional awareness. I think the novelty of this paper is good. But my presentation is poor and the experiments are so insufficient.

**Strengths:**

1. The novelty seems good, a very interesting method to combine the MIM and CL together;
2. Many analysis experiments enhance the quality of this paper;

**Weaknesses:**

1.I do not think attention response (figure 1&7&8) can represent the the quality of the learned features. For example, the attention responses of models using the CLIP model as a supervisory signal all appear poor, but their actual performance is often much better.
2.It seems that the authors did not use \citep{} correctly while writing and instead used \cite{}. This resulted in an inconsistent citation format throughout the entire paper.
3.I suggest the authors to provide a comparison on training cost, because the proposed method seems have larger computational cost.
4.Experimental results are insufficient: longer epochs and larger models (L-scale) need to be validated, which is crucial for self-supervised learning. Downstream task results are also needed, such as semantic segmentation and object detection. (This is the biggest concern!)

**Questions:**

Do you think the downstream task performance is good enough to evaluate a self-supervised model? If so, the current MIM methods have achieved very good results and I think the proposed method has NO potential to reach such high results. If not, what metrics is needed do you think to evaluate a pretrained model?

---

> ### Author Response · Authors · 2024-11-23
> **Response to Reviewer Efcx (1/2)**
>
> We appreciate your thoughtful feedback and are delighted to see the novelty of this work being acknowledged. The response to your questions are shown below.
>
> > I do not think attention response (figure 1&7&8) can represent the the quality of the learned features. For example, the attention responses of models using the CLIP model as a supervisory signal all appear poor, but their actual performance is often much better.
>
> We agree that the quality of attention maps is not the golden metric of feature quality. The attention visualization results in the manuscript are intended to assist in interpreting the analysis of CL & MIM. The attention difference between SemanticMIM and vanilla MIM intuitively reflects the impact on capturing object semantics by introducing compression. We believe that the analysis is the key finding that we aim to convey in this work. By visualization, we seek to explore the properties of the proposed method as thoroughly as possible for better understanding. Besides, we also provide evaluation results of SemanticMIM on several downstream tasks, following commonly used evaluation protocols in the self-supervised learning field. These results quantitatively measure the improvements in feature quality brought by SemanticMIM indirectly but objectively. Our method demonstrates significant advantages under linear probing protocols. We hope these results provide evidence for the effectiveness of SemanticMIM.
>
> > It seems that the authors did not use \citep{} correctly while writing and instead used \cite{}. This resulted in an inconsistent citation format throughout the entire paper.
>
> We apologize for our oversight and have corrected all the citation formats.
>
> > I suggest the authors to provide a comparison on training cost, because the proposed method seems have larger computational cost.
>
> We have provided analysis of the training cost in supplementary material. For your convenience, we list a table comparing the relative training cost between the baseline and SemanticMIM. Suppose the input image is 224x224, patch size is 16, thus the input token sequence length is 196. As shown in the table below, our method has a training cost similar to the baseline. This is because, although we have added an additional block per layer, each block only processes a subset of tokens, and only the proxy tokens are doubly computed.
>
> | Setting | ViT-B | ViT-L |
> | --- | --- | --- |
> | Baseline | 100% |100% |
> | proxy=1 | 93.3% | 93.7%|
> | proxy=2 | 93.9% | 94.3%|
> | proxy=4 | 95.0% | 95.4%|
> | proxy=8 | 97.2% | 97.6%|
> | proxy=16 | 101.7% | 102.1%|
> | proxy=32 | 110.6% | 111.0%|
>
> > Do you think the downstream task performance is good enough to evaluate a self-supervised model? If so, the current MIM methods have achieved very good results and I think the proposed method has NO potential to reach such high results. If not, what metrics is needed do you think to evaluate a pretrained model?
>
> Thank you for raising this insightful question. We believe it is an important topic for all researchers in the self-supervised learning field to consider. Currently, using downstream task performance as evaluation metric does indeed have many limitations. Each vision task can only reflect certain aspects of feature properties, and the results are heavily influenced by finetuning recipes and hyperparameters. In our view, an ideal evaluation method should be unsupervised and zero-shot to examine the real generalization ability. However, due to the lack of a unified framework for vision tasks, achieving such an ideal evaluation framework remains challenging. As a compromise, evaluation protocols that do not require additional finetuning, such as linear probing classification and unsupervised segmentation, impose higher demands on the feature quality and guarantee that all capabilities come purely from pretraining. Our proposed method not only excels in terms of interpretability but also shows significantly better performance compared to baselines under similarly challenging evaluation protocols. Exploring comprehensive and accurate evaluation metrics for pretrained methods is a valuable direction for future research.

---

> ### Author Response · Authors · 2024-11-23
> **Response to Reviewer Efcx (2/2)**
>
> > Experimental results are insufficient: longer epochs and larger models (L-scale) need to be validated, which is crucial for self-supervised learning. Downstream task results are also needed, such as semantic segmentation and object detection.
>
> Following your suggestions, we added experiments as below.
>
> ### Semantic Segmentation
> The results of semantic segmentation are provided in Tab.1 in the manuscript.
>
>
> ### Object detection & Instance segmentation
>
> | Method | Seg (mAP) | Bbox (mAP) |
> |--------------|-----------|------------|
> | Baseline | 40.6 | 46.6 |
> | SemanticMIM |  41.8      |  47.6 |
>
> We evaluate the ViT-Base model pretrained by BEiT and SemanticMIM on the COCO 2017 dataset. Following the recipe in Swin Transformer[1], we adopt Cascade Mask R-CNN[2] as framework and train for 36 epochs with multi-scale training. Since ViT does not have hierarchical architecture to reduce the feature map resolution, we remove the last four layers of the pretrained model to avoid OOM. The results are shown in the table above. SemanticMIM brings +1.0 box AP and +1.2 mask AP against the baseline, demonstrating the effectiveness of our proposed method.
>
> ### Model size scaling
>
> | Method | Linear Probing | Finetuning |
> |--------------|-----------|------------|
> | ViT-S | 44.3 | 81.2 |
> | ViT-B | 59.7 | 83.6 |
> | ViT-L | 64.6 | 84.5 |
>
> Following your suggestions, we conducted experiments on different model sizes. We use MaskFeat training framework and apply SemanticMIM during pretraining. The training and evaluation recipes of these experiments remain the same as detailed in our manuscript except for the model architecture. During evaluation, we only feed the classifier with the average pooling of patch tokens, named Patch in our manuscript. The results are shown in the table above. It can be observed that the performance of the model improves as the number of parameters increases. This is particularly evident in the performance under linear probing protocol, where ViT-L shows a clear lead. At the same time, we observed that the linear probing performance of ViT-S is significantly reduced compared to ViT-B. We think this is because our proposed method increases the difficulty of the task through compression. For shallower models, high task difficulty will lead to insufficient learning. Adjusting the task difficulty by setting appropriate number of proxy tokens should be able to fully release the capabilities of models of different sizes.
>
> [1] Swin transformer: Hierarchical vision transformer using shifted windows. ICCV 2021.
> [2] Cascade R-CNN: Delving into high quality object detection. CVPR 2018.

---

> ### Comment · Reviewer_Efcx · 2024-11-26
> **Thanks for the responses**
>
> I still believe that attention response can not used for evaluating the goodness of a method. I strongly suggest that the authors note that in the paper: it is so unfair to compare the attention responses with BEiT.
>
> Overall, this paper is unlikely to have a significant positive impact in the MIM field, as its performance potential is almost negligible compared to state-of-the-art MIM methods. However, based on the authors' proactive responses, I choose to raise my score from reject to weak reject.
>
> TO AC: I believe this paper is overall below the acceptance standard of ICLR. If you wish to accept it, I think it is acceptable, but my recommendation is to reject.

---

### Official Review · Reviewer_H81S · 2024-10-31

**Soundness:** 3
**Presentation:** 3
**Contribution:** 3
**Rating:** 6
**Confidence:** 3

**Summary:**

The paper introduces SemanticMIM, a framework that combines Masked Image Modeling (MIM) and Contrastive Learning (CL) for enhanced visual representation.

Key Contributions:

Theoretical Insights: It analyzes the complementary strengths of MIM and CL, emphasizing their different approaches to semantic modeling—CL focuses on global semantics, while MIM emphasizes local details.

Framework Design: SemanticMIM integrates CL's benefits within the MIM framework using a proxy architecture that combines compression and reconstruction processes into a single learning framework.

Experimental Results: The framework demonstrates significant performance improvements in distinguishing specific object semantics and identifying relevant features, outperforming both MIM and CL in various tasks.

Conclusion: SemanticMIM effectively captures global and spatial information, leading to notable advancements in visual representation for downstream applications.

**Strengths:**

The strengths of this paper lie in its innovative integration of Masked Image Modeling (MIM) and Contrastive Learning (CL) within a unified framework, effectively leveraging the advantages of both approaches. Additionally, SemanticMIM excels at capturing both global semantics and local features, enhancing the ability to distinguish specific object semantics. Finally, in addition to quantitative experiments, the paper also includes extensive qualitative analyses and visualizations to demonstrate the effectiveness of the framework.

**Weaknesses:**

The paper's limitations include insufficient quantitative experiments, particularly a lack of tests with models of varying sizes across different settings. Additionally, while the paper provides segmentation results for downstream tasks, the promising attention effects raise expectations for object detection performance as well. It would be beneficial for the authors to include more experimental results in the rebuttal stage.

**Questions:**

The shortcomings in the experimental section raise concerns about whether this training method can scale to larger models and be applied to a broader range of downstream tasks. If the authors can provide additional experimental results, I would be willing to reconsider and potentially increase the score.

---

> ### Author Response · Authors · 2024-11-23
> **Response to Reviewer H81S**
>
> Thank you for your thoughtful review and helpful advice. We added some experiments as below.
> ### Object detection & Instance segmentation
>
> | Method | Seg (mAP) | Bbox (mAP) |
> |--------------|-----------|------------|
> | Baseline | 40.6 | 46.6 |
> | SemanticMIM |  41.8      |  47.6 |
>
> We evaluate the ViT-Base model pretrained by BEiT and SemanticMIM on the COCO 2017 dataset. Following the recipe in Swin Transformer[1], we adopt Cascade Mask R-CNN[2] as framework and train for 36 epochs with multi-scale training. Since ViT does not have hierarchical architecture to reduce the feature map resolution, we remove the last four layers of the pretrained model to avoid OOM. The results are shown in the table above. SemanticMIM brings +1.0 box AP and +1.2 mask AP against the baseline, demonstrating the effectiveness of our proposed method.
>
> ### Model size scaling
>
> | Method | Linear Probing | Finetuning |
> |--------------|-----------|------------|
> | ViT-S | 44.3 | 81.2 |
> | ViT-B | 59.7 | 83.6 |
> | ViT-L | 64.6 | 84.5 |
>
> Following your suggestions, we conducted experiments on different model sizes. We use MaskFeat training framework and apply SemanticMIM during pretraining. The training and evaluation recipes of these experiments remain the same as detailed in our manuscript except for the model architecture. During evaluation, we only feed the classifier with the average pooling of patch tokens, named Patch in our manuscript. The results are shown in the table above. It can be observed that the performance of the model improves as the number of parameters increases. This is particularly evident in the performance under linear probing protocol, where ViT-L shows a clear lead. At the same time, we observed that the linear probing performance of ViT-S is significantly reduced compared to ViT-B. We think this is because our proposed method increases the difficulty of the task through compression. For shallower models, high task difficulty will lead to insufficient learning. Adjusting the task difficulty by setting appropriate number of proxy tokens should be able to fully release the capabilities of models of different sizes.
>
> [1] Swin transformer: Hierarchical vision transformer using shifted windows. ICCV 2021.
> [2] Cascade R-CNN: Delving into high quality object detection. CVPR 2018.

---

> > ### Comment · Reviewer_H81S · 2024-11-24
> > **Response to Authors**
> >
> > Thank you to the authors for their patient responses. However, the experimental results presented here do not resolve my concerns, as the core issue lies in the experimental setup.
> >
> > Regarding downstream tasks, the Out-Of-Memory (OOM) problem with plain Vision Transformer backbones in such tasks has been widely discussed. For example, ViTDet[1] achieves a box mAP of 51.2 and a mask mAP of 45.5 on COCO using ViT-B, which significantly surpasses the experimental results provided by the authors in their responses.
> >
> > As for model size, a more reasonable approach would be to directly compare the performance of different MIM methods under linear probing and fine-tuning. Unfortunately, the authors did not include a comparison with existing methods. Taking the classic MIM work MAE as an example, the results from the MAE GitHub repository show fine-tuning performance on ImageNet-1K as ViT-B: 83.6 and ViT-L: 85.9, which are not inferior to the proposed method. Admittedly, the performance gap here may be attributed to differences in training settings, but it could also indicate that the proposed method itself is not effective.
> >
> > Therefore, I believe the authors need to conduct a more comprehensive investigation in the experimental section to present more reliable results.
> >
> > 1. Exploring Plain Vision Transformer Backbones for Object Detection

---

> > > ### Author Response · Authors · 2024-11-25
> > > **Response to Reviewer H81S**
> > >
> > > Thank you for your timely response.
> > >
> > > First, we appreciate your suggestion for better settings in object detection experiments.
> > > ViTDet is undoubtedly better than the detection framework we are currently using, especially considering that we reduce layers which hurts the performance.
> > > We will try this more reasonable setting and rerun the experiments, although this may take some time and cannot be completed during the discussion period.
> > > Although the current detection setup is not optimal, we believe that the relative performance improvement can still serve as a reference for the effectiveness of our method.
> > >
> > > As for model size experiments, directly comparing with SOTA methods is a compelling way. However, different baseline settings are inherently divergent, especially considering that our training epochs are much fewer than those of official implementations, making performance comparisons unfair.
> > > Notably, the primary goal of our experiments is to establish a clean, simple, and fair baseline to substantiate our claim, but not to train a model that achieves optimal performance.
> > > The improvement brought by our method over the baseline is significant, especially in the challenging setting of linear probing, which we think is the key to validate the effectiveness of introducing compression into MIM.
> > >
> > >
> > > On the other hand, the [PROXY] structure of SemanticMIM actually serves as a plugin, adaptable to any method following the MIM framework. Therefore, considering that methods within the same framework have similar properties, similar performance improvements can be expected when switching different baselines, such as from BEiT to MAE, which makes it less significant for exploring the effectiveness of SemanticMIM.

---

> > > > ### Comment · Reviewer_H81S · 2024-11-27
> > > >
> > > > Thank you for your response. I appreciate the insightful analysis and the strong attention map visualizations that highlight the potential of the proposed MIM method, which was the reason I initially gave a weak accept. However, after thorough discussion, I believe the experimental setup has significant shortcomings that fail to convincingly demonstrate the effectiveness of the method. As a result, my overall evaluation has been slightly adjusted downward.

---

### Official Review · Reviewer_zqBB · 2024-11-04

**Soundness:** 3
**Presentation:** 2
**Contribution:** 3
**Rating:** 6
**Confidence:** 3

**Summary:**

This paper first compares the pros and cons of two related but different lines of work: masked image modeling (MIM) and contrastive learning (CL). They then propose SemanticMIM which brings the pros of CL, i.e. suppression, and global representation, etc, into MIM. Experiments show improvement in several tasks.

**Strengths:**

1. The analysis of MIM and CL can be beneficial to the community.
2. Introducing [PROXY] tokens between [CLS] and [MASK] as the solution is simple.
3. Visualization in the experiments highlights the claims of the paper.

**Weaknesses:**

1. The presentation is not good. The figures do not help explain the methods.

**Questions:**

1. The layout of Fig. 1 can be changed to fit its goal of comparing MIM and CL as the following.
Image A BeiT-A MoCov-A Ours-A
Image B BeiT-B MoCov-B Ours-B
The current layout is confusing at first glance since A and B are "up vs. down" on the left subfigure but "left vs. right" on the right subfigure.

2. Fig. 1 needs a clearer caption to explain it. The caption does not explain what the reader should be observing here. In what way is "ours" better than the baselines? The paper only explains it later in L49, L246, and L264. Moreover, in L49, how does a reader realize the following claim "MIM focuses on the reconstruction of partially corrupted images, serving as a pretext task that facilitates the model’s ability to infer local patterns from contextual information rather than grasping global semantics" from the figure? The color yellow for the boxes is too similar to the chosen colormap.

3. Fig. 2 can be also made clearer. What is the purpose of the target generators here? What is being trained? The [CLS] tokens and [MASK] tokens should be indicated in this figure. The token color of Fig.2 and Fig.3 should match.

4. In Fig 3, why does the [MASK] token in MIM go straight to [TARGET]? As in Fig. 2, same as CL, MIM also outputs some tokens (blue). The major difference of [MASK] token having positional embedding should be indicated in Fig.2 or 3.

5. In L262, there is a missing space between SimMIM and the citation.

6. In Fig. 4, it does not seem like compression for SemanticsMIM since the number of [MASK] matches the number of [IMG].

7. In L344, semanticMIM -> SemanticMIM.

8. The baselines seem outdated (2021~2022). Can SemanticMIM be compared to [1] or [2]?

9. In the experiment section, why use the term  "[CLS] token with i.e. [PROXY] token" but not just "[PROXY] token"? It is confusing to read.

10. I think the caption in Fig.5 and Fig. 6 should be y-axis (singular).

11. In Sec. 4.4, as a comparison, how many [IMAGE] tokens are there?

12. In Tab. 1, Why compare to only Linear for PascalVOC and only FT for ADE20K? Is using the CLS and Patch tokens as auxiliary inputs for the classifier necessary in the ImageNet experiments? Why is there no "using CLS or Patch tokens" for PascalVOC and ADE20K?

[1] MIMIC: Masked Image Modeling with Image Correspondences. CVPRW 2024.
[2] Learning Vision from Models Rivals Learning Vision from Data. CVPR 2024.

---

> ### Author Response · Authors · 2024-11-23
> **Response to Reviewer zqBB (1/2)**
>
> Thank you for your detailed and helpful advice. Following your suggestions, we have revised our manuscript, especially figures, and highlighted the changes in red for your convenience.
>
> Our response to your questions is set out below:
>
> >The layout of Fig. 1 can be changed to fit its goal of comparing MIM and CL as the following. Image A BeiT-A MoCov-A Ours-A Image B BeiT-B MoCov-B Ours-B The current layout is confusing at first glance since A and B are "up vs. down" on the left subfigure but "left vs. right" on the right subfigure.
>
> We have rearranged the content in Fig.1 following your advice and separated two samples apart to make it clear.
>
> > Fig. 1 needs a clearer caption to explain it. The caption does not explain what the reader should be observing here. In what way is "ours" better than the baselines? The paper only explains it later in L49, L246, and L264. Moreover, in L49, how does a reader realize the following claim "MIM focuses on the reconstruction of partially corrupted images, serving as a pretext task that facilitates the model’s ability to infer local patterns from contextual information rather than grasping global semantics" from the figure? The color yellow for the boxes is too similar to the chosen colormap.
>
> We have rewritten the caption with a brief description of the phenomenon revealed in Fig.1, labeled the query patches with solid red boxes to make them prominent, and pointed out the key property of MIM for better understanding. The caption is now changed to "Attention response of different self-supervised vision transformers. The queries are marked with red boxes. MoCov3 fails to follow the query and BEiT focuses too much on neighboring patches, while SemanticMIM distinguishes different objects and approximates their segmentation masks. MoCov3 and BEiT show the result from $10^{th}$ layer while Ours are from $8^{th}$ layer."
>
> > Fig. 2 can be also made clearer. What is the purpose of the target generators here? What is being trained? The [CLS] tokens and [MASK] tokens should be indicated in this figure. The token color of Fig.2 and Fig.3 should match.
>
> The target generator here refers to the module that transfers the input image to the groundtruth used for training. For example, it is a dVAE in BEiT, a momentum encoder in MoCo, and an identity function in SimMIM. Both the target generator and augment operators are offline modules that do not update via gradient. The vision transformer and prediction head are modules being trained. We have added description to the figure caption. Besides, the token colors in Fig.2.3.4 are now matched.
>
> > In Fig 3, why does the [MASK] token in MIM go straight to [TARGET]? As in Fig. 2, same as CL, MIM also outputs some tokens (blue). The major difference of [MASK] token having positional embedding should be indicated in Fig.2 or 3.
>
> In the original version of Fig.2, the blue tokens refer to the output of corresponding yellow tokens. So in CL, the output token that goes to the target is [CLS] token, while in MIM, [MASK] tokens. We have realized that such color arrangement can be confusing. Therefore, we have modified the token colors in Fig.2 and Fig.3, with each color corresponding to a different kind of token. Also, thanks for your advice and we have marked the positional embedding of each token in Fig.3 for clarity.
>
> > In L262, there is a missing space between SimMIM and the citation.
>
> Thanks for pointing out the typo, we've fixed it.
>
> > In Fig. 4, it does not seem like compression for SemanticsMIM since the number of [MASK] matches the number of [IMG].
>
> We have redrawn Fig.4, using different numbers of tokens to represent compression and aligned the token colors with Fig.2 and Fig.3 and emphasize this in the caption.
>
> > In L344, semanticMIM -> SemanticMIM.
>
> Thanks for pointing out the typo, we've fixed it.
>
> > The baselines seem outdated (2021~2022). Can SemanticMIM be compared to [1] or [2]?
>
> MIMIC builds a new dataset that contains multi-view data. SynCLR uses LLM and T2I models to synthesize text-image pairs. Both of these works aim to improve model performance from the data level, while the goal of our work is to analyze the properties of the training framework and improve the semantic extraction ability from the model structure perspective. Therefore, we believe that the motivations and improvements of SemanticMIM and these two works are orthogonal.
>
> > In the experiment section, why use the term "[CLS] token with i.e. [PROXY] token" but not just "[PROXY] token"? It is confusing to read.
>
> "[CLS] token i.e. [PROXY] token" refers to the fact that [PROXY] tokens are learnable embeddings and play the same role of [CLS] token in a vanilla transformer during inference. We have realized that it does confuse people and revised it in the experiment section.
>
> > I think the caption in Fig.5 and Fig. 6 should be y-axis (singular).
>
> Thanks for pointing out the typo, we fixed it and moved the captions to the main text.

---

> ### Author Response · Authors · 2024-11-23
> **Response to Reviewer zqBB (2/2)**
>
> > In Sec. 4.4, as a comparison, how many [IMAGE] tokens are there?
>
> The resolution of the input image is 224x224 and patch size is 16, so the number of [IMG] tokens is (224/14)^2=196. We have added this to the manuscript for clarity.
>
> > In Tab. 1, Why compare to only Linear for PascalVOC and only FT for ADE20K? Why is there no "using CLS or Patch tokens" for PascalVOC and ADE20K?
>
> We followed the evaluation protocol in BEiT[1] of segmentation on ADE20k. However, BEiT does not provide a linear probing evaluation protocol for the segmentation task, which is more difficult than fine-tuning and can directly reflect the quality of pre-trained features because the model is frozen. Therefore, we follow the linear evaluation method of Leopart[2], whose evaluation protocol is performed on PascalVOC dataset.
>
> > Is using the CLS and Patch tokens as auxiliary inputs for the classifier necessary in the ImageNet experiments?
>
> Using more features leads to better accuracy in classification tasks. BEiT uses combination of the [CLS] token and patch tokens from three layers. In our experiment, we didn't use this trick and evaluated separately to see the impact of our methods on different kinds of tokens.
>
> [1] BEiT: BERT Pre-Training of Image Transformers. ICLR 2022.
> [2] Self-supervised learning of object parts for semantic segmentation. CVPR 2022.

---

> > ### Comment · Reviewer_zqBB · 2024-11-25
> >
> > The authors have addressed most of my concerns. However, I still think Fig. 3 and Fig. 4 can have more detailed captions for explanation—especially Fig. 3. It is the most important figure as it expresses the high-level idea of this work. For example, in the caption you can explain why using both "Compression" and "Reconstruction" is better than using either one of them for your task.

---

> > > ### Author Response · Authors · 2024-11-26
> > > **Response to Reviewer zqBB**
> > >
> > > Thanks for your further advices. We have added explanations to the captions of Fig.3 and Fig.4 as below and highlighted the modified part in purple in the manuscript.
> > >
> > > Fig.3 caption:
> > >
> > > Information propagation of the contrastive learning, masked image modeling, and our proposed SemanticMIM. Numbers indicates position ids and the slash means position-irrelevant. The compression structures present in CL methods endow the model with a better ability to capture semantics. Inspired by this, we introduce similar compression structures into MIM, aiming to enhance the semantic awareness on top of the original positional awareness of MIM.
> > >
> > > Fig.4 caption:
> > >
> > > Comparison of the architecture. MIM only focuses on Reconstruction. In SemanticMIM, since the number of [PROXY] is much smaller than that of [IMG] , information is compressed first (left) and then transmitted to [MASK] to complete the reconstruction (right). This design introduces compression while remaining compatible with the original MIM framework.

---

### Author Response · Authors · 2024-11-23
**Response to All Reviewers**

We appreciate all reviewers for their careful reading and insightful feedback and we are pleased to see that all reviewers found our proposed analysis and method to be novel, insightful, and beneficial to the community. The main concerns raised by the reviewers focused on presentation and experiments. We have addressed these questions in detail, revised the manuscript accordingly, and conducted additional experiments. Here is a brief summary:

1. We have added experiments on scaling up the model and serving as backbone for object detection (see response to Reviewer H81s and Efcx).
2. We have redrawn all the figures to enhance the quality of the presentation.
    - Fig.1: We adjusted the layout of the images, used red solid boxes to annotate query image patches. We rewrite caption with descriptions of the model properties emerged for better understanding. We also pointed out the layers corresponding to the attention responses for clarity.
    - Fig.2: We modify the token color, matched with Fig.3 and Fig.4 to make it less confusing and mark the query tokens.
    - Fig.3: Token colors are matched and position ids are added to each kind of tokens.
    - Fig.4: We match the token colors, add target tokens and adjust the token numbers to emphasize compression.
3. We have corrected the citation format errors and some typos in the manuscript.
4. We clarified the concept of CL and MIM in our manuscript to make them more rigorous.
5. In the experimental section, we included more implementation details of the modification of SemanticMIM compared to general MIM framework.

Thanks again for all reviewer's comments and acknowledgment of our work. We sincerely look forward to the opportunity to engage in further discussions.

---

### Meta-Review · Area_Chair_K2YC · 2024-12-17

**Metareview:**

This paper presents a visual pre-training algorithm that unifies masked image modeling and contrastive learning. It first formulates these two types of pre-training methods using one framework, and then proposes a new pipeline that uses compressed image tokens to guide the model to reconstruct the masked tokens. Although with a smaller number of [PROXY] tokens, the overall framework is **not** essentially different from the vanilla MIM -- note that MIM, with a large masking ratio, can force the model to work on highly-compressed image tokens. Also, the experiments are not strong enough to validate the effectiveness; in particular, experiments on larger backbones and downstream tasks are not complete.

The overall rating of this paper falls on the borderline (5/5/6/6). The reviewers are mostly concerned about the writing (some typos, missing technical details, etc.) and experiments (small models, short training schedules, weaker baselines, etc.). Given this many points to resolve, the AC cannot suggest acceptance this time and suggests the authors to continue improving this work.

**Additional Comments On Reviewer Discussion:**

All reviewers responded after the rebuttal. Two (slightly) negative reviewers insisted on rejection, and one positive reviewers downgraded the recommendation from clear acceptance to slight acceptance.

---

### Decision · Program_Chairs · 2025-01-22

Reject